# Genetic Diversity Assessment of *Cupressus gigantea* W. C. Cheng & L. K. Fu Using Inter-Simple Sequence Repeat Technique

**Ximei Ji** [1,2,†], **Yaxuan Jiang** [2,†], **Jianxin Li** [2], **Pei Lei** [1,*] **and Fanjuan Meng** [1,*]

1. Jilin Provincial Key Laboratory of Tree and Grass Genetics and Breeding, College of Forestry and Grassland Science, Jilin Agricultural University, Changchun 130118, China; 18800465846@163.com
2. College of Life Science, Northeast Forestry University, Harbin 150040, China; jiangyaxuan1206@163.com (Y.J.); 15776685360@163.com (J.L.)
* Correspondence: lppaper@163.com (P.L.); mfjtougao@163.com (F.M.)
† These authors contributed equally to this work.

**Abstract:** *Cupressus gigantea* W. C. Cheng & L. K. Fu is an endemic conifer tree species that is distributed widely along the northern portion of the deep gorge of the Yarlung Tsangbo River on the Tibetan Plateau. However, as a key plant species growing on the Tibetan plateau, *C. gigantea* has since become an endangered species due to habitat loss and degradation, overexploitation, and other factors. It has been listed as a first-grade national protected wild plant species in China. Accordingly, to conserve this plant species, we should obtain more information on its genetic structure. In this study, the genetic diversity and structure among 67 samples were evaluated by the inter-simple sequence repeat (ISSR) technique. Overall, 78 bands were produced with a molecular length of 200 bp to 3100 bp using 10 ISSR primers. The mean values for the average number of alleles (*Na*), effective number of alleles (*Ne*), Nei's gene diversity (*H*), and Shannon's information index (*I*) were 1.529, 1.348, 0.199, and 0.293, respectively. Additionally, the number of polymorphic loci (NPLs) and percentage of polymorphic loci (*PPLs*) averaged 41.25 and 52.90, respectively. Further, total variation among populations was 14.2%, while that within populations was 85.8%; accordingly, the within-population genetic differentiation was found to be significant (*p* < 0.001). These results demonstrated that a genetic structure model with *K* = 3 fitted the data best, which agreed with the unweighted pair group method with arithmetic average (UPGMA) cluster and the principal coordinate analysis (PCoA). These findings are beneficial for ensuring the development and genetic protection of *C. gigantea* populations in the future.

**Keywords:** *Cupressus gigantea* W. C. Cheng & L. K. Fu; genetic diversity; inter-simple sequence repeat; unweighted pair group method with arithmetic averages; conservation measures

## 1. Introduction

*Cupressus gigantea* W. C. Cheng & L. K. Fu (Tibetan juniper), *Cupressaceae*, *Cupressus*, is an endemic plant species [1]. The number of chromosomes in *C. gigantea* somatic cells is 2n = 22, and it mainly reproduces from seeds [2,3]. *C. gigantea* is sparsely distributed along the northern part of the deep gorge of the Yarlung Tsangbo River on the Tibetan Plateau, also known as the Yarlung Zangbo River cypress [4]. This region features extremely severe environmental conditions, including high radiation, cold temperature, strong winds, and barren soil [5]. Nevertheless, *C. gigantea* can survive in this harsh habitat where it plays a pivotal role in soil protection and the prevention of desertification [6,7]. Additionally, this evergreen tree is relatively tall (up to 30–50 m in height) and lives long (from 100 to 1000 years), making *C. gigantea* one of the long-lived endemic cypresses in China [8–10]. Therefore, it is generally respected as being a god tree by local residents [11]. At the same time, it is also an important raw material for making Tibetan incense, so *C. gigantea* has important cultural value for the local residents [12]. It is also well known for its ornamental and medicinal properties [13]. Notably, *C. gigantea* has high economic value

due to its high wood density, straight grain, and radial symmetry [10]. Up to now, studies on *C. gigantea* have mainly focused on its total protein extraction, determination of the complete chloroplast genome, comprehensive transcriptome characterization, community characteristics, growth characteristics, propagation, and other related aspects [3,14–17]. To date, the analysis of the genetic population of *C. gigantea* is lacking.

However, as a key plant species growing on the Tibetan plateau, *C. gigantea* has since become an endangered species due to a low seed-setting rate, habitat loss, and degradation, as well as extensive logging for Tibetan incense production and timber production [18–22]. Currently, *C. gigantea* has been listed as a first-grade national protected wild plant species in China [17,23]. Hence, conservation efforts must be initiated to halt the decline of this tree species and the associated loss of biodiversity. Generally, in this respect, an evaluation of the genetic structure and diversity of *C. gigantea* is highly important for their future germplasm resource management and the formulation of breeding strategies. Moreover, studying the genetic structure in *C. gigantea* is fundamental for understanding the internal variation in the species, which is conducive to the identification and preservation of germplasm. While traditional morphological and biochemical characteristics are often used for this purpose. However, their limitations in assessing genetic diversity highlight the need for more precise and accurate markers. In recent years, a series of molecular marker techniques have been used to analyze the genetic structures of various plant species [24].

To date, various molecular marker techniques developed over nearly four decades have led to insights into the biology, genetics, and genome evolution of conifers [25]. Accordingly, to estimate the genetic diversity in *C. gigantea* populations, some DNA-based marker techniques have been carried out, such as amplified fragment length polymorphism (AFLP), inter-simple sequence repeat (ISSR), randomly amplified polymorphic DNA (RAPD), and polymorphic fluorescent-labeled SSR markers. Among them, in 2008, Zha reported that nine populations of *C. gigantean* were sampled and analyzed by AFLP markers, indicating high genetic diversity in the populations and significant genetic variation within the various populations [12]. Another group discovered that the ISSR primers produced more polymorphic bands than the RAPD markers. Additionally, there were high levels of genetic differentiation within populations ($p < 0.001$) [26]. Similarly, the AMOVA of SSR showed that genetic variation mainly existed within populations [27]. Overall, there are still some inadequacies in the aforementioned studies. On the one hand, the collection time is 20 years ago, and the populations of *C. gigantea* have undergone dramatic changes due to the lack of effective conservation measures. On the other hand, the individual studies collected only five groups, which lacks universality. Therefore, our research on the existing populations of *C. gigantea* is more conducive to the development of more detailed conservation measures.

However, some molecular markers have shown some shortcomings including the high cost of AFLP and the radioactive labeling of RFLP [28]. Until now, SSR markers for mining SSR motifs in the whole genome greatly reduce the cost but show somewhat lower consistency with dominant marker data, and some plants' genomic data are not published [29]. ISSR markers have the advantages of simplicity, high reproducibility, reliability, and stability, as well as important applications in genetic diversity research, gene mapping, and genetic fingerprints [30]. The ISSR markers are specific by using longer primers based on SSRs. They also have a higher annealing temperature, which helps generate clearer and more consistent amplification [31]. In addition, this technique can also evaluate the genetic relationship between accessions and enables the construction of genetic linkage maps [32]. Notably, this method is very efficient in terms of time and labor when seeking to compare genetic resources from different collection sites [33,34]. Abundant evidence suggests that ISSR markers have proven to be a powerful tool for detecting genetic differentiation and describing the germplasm of various cultivated and wild plant species [35,36]. These characteristics make ISSR markers particularly suitable for assessing the genetic structure in *C. gigantea*. Surprisingly, however, this type of molecular marker has not been used in recent years to investigate the genetic variation in *C. gigantea*.

In this study, we screened 10 ISSR primers to uncover the genetic relationship and genetic diversity of eight *C. gigantea* populations collected along the bank of the Yarlung Tsangbo River. Using this information, we sought to comprehensively reveal the genetic variation level among and within populations of *C. gigantea*, and thereby proposing effective conservation strategies for *C. gigantea*.

## 2. Materials and Methods

### 2.1. Materials

Sixty-seven *C. gigantea* samples were collected along the coast of the Yarlung Tsangbo River of Tibet. The detailed information of the collection sites of all samples used in this study is listed in Figure 1. Fresh and young leaves of all the samples were collected in the Spring season (March) of the year 2017 and preserved at the Genetic Department of College of Life Science of Northeast Forestry University, Harbin, China.

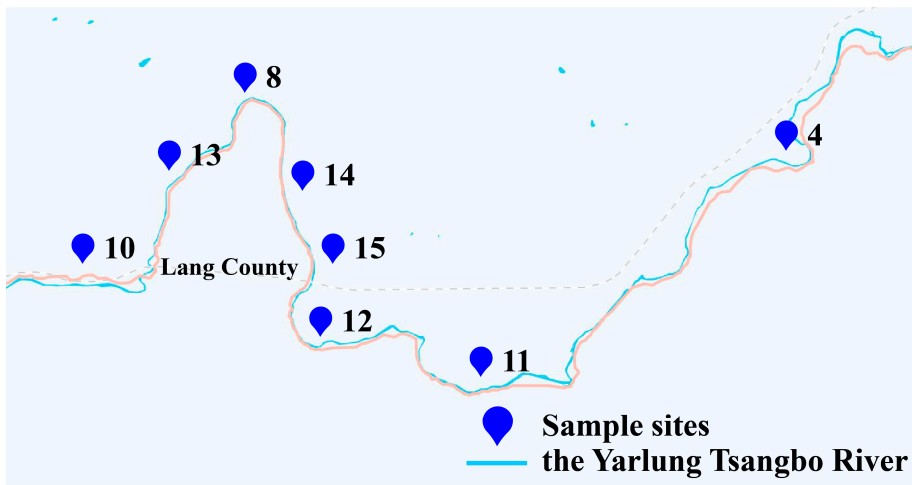

**Figure 1.** Collection sites of eight populations from the Yarlung Tsangbo River of Tibet: the numbers represents the collection sites of sample; the red line represents China's Yaya highway.

### 2.2. DNA Extraction

Fresh and young leaves were frozen in liquid nitrogen for DNA extraction. Total DNA was extracted from 0.1 g of *C. gigantea* leaves using the modified CTAB method [37]. DNA samples were estimated qualitatively and quantitatively by a spectrophotometer and 0.8% agarose gel electrophoresis, respectively. Then, a 100 ng·$\mu$L$^{-1}$ dilution of DNA samples was prepared for further PCR analysis.

### 2.3. ISSR Analysis

ISSR amplification was carried out according to a previous method, with some modifications described by Xie et al. [38]. Fifty ISSR primers (University of British Columbia, Vancouver, BC, Canada) were tested, and ten ISSR primers (UBC #808, #824, #827, #836, #841, #842, #847, #856, #857, and #873) showed clear polymorphic bands and good repeatability, and these were screened further using genetic analysis. All primers were synthesized in Shanghai Sangon Biological Engineering Technology and Service (Co., Ltd., Shanghai, China). The ISSR reactions were amplified in total mixture volume of 20 $\mu$L containing DNA template (80 ng), 1 $\mu$L primer (10 mM$^{-1}$), 2 $\mu$L *dNTP*s; (0.5 mM·L$^{-1}$), 2.5 $\mu$L 10 × PCR reaction buffer, 2.5 $\mu$L MgCl$_2$ (2.5 mM·L$^{-1}$), 0.5 $\mu$L Taq DNA polymerase (5 U·$\mu$L$^{-1}$) (Takara Biotechnology Co., Ltd., Dalian, China), and 9.5 $\mu$L ddH$_2$O.

PCR cycling conditions were carried out in an Eppendorf PCR an Amplifier GeneAmp PCR System 9700 (Eppendorf, Hamburg, Germany) with the following protocol: run at 94 °C for 5 min, followed by 40 cycles of 94 °C for 30 s, 55 °C for 45 s, and 72 °C for 2 min, and a final extension at 72 °C for 10 min. To estimate the sizes of DNA fragments, a 2000 bp DNA weight marker (NewEngland BioLabs, Massachusetts, USA) was included.

The amplified products were separated in 2.0% agarose gels stained with GelRed ($0.5\ \mu L\cdot mL^{-1}$) and imaged under an ultraviolet illuminator and photographed.

### 2.4. Statistical Analysis

For each ISSR marker, all the assessed bands were distinguished based on the presence (1) or absence (0) of genetic diversity. Based on distinguished results, a binary data matrix was obtained. Accordingly, a dendrogram was constructed according to the unweighted pair group method with arithmetic average (UPGMA) of NTSYS-pc version 2.10e [39]. The genetic parameters including *Na*, *Ne*, *H*, *I*, *NPLs*, and *PPLs* were calculated by the POPGENE (VERSION-1.31) software package [40]. Genetic structure was constructed by STRUCTURE 2.3.4 Microsoft [41]. Genetic differentiation was assessed by the method of molecular variance (AMOVA) version 1.55 [42].

## 3. Results

### 3.1. ISSR Analysis

We successfully screened 10 primers to amplify all the collected samples (Table 1). Altogether, these screened ISSR markers produced a total of 78 bands from UBC-836 (4) to UBC-857 (13), with molecular lengths of 200–3100 bp. Among them, 67 polymorphic bands were amplified. The percentage of polymorphism varied from 50% (UBC-836) to 100%, with an average of 85.90%. Three ISSR markers (UBC-824, UBC-827, and UBC-847) exhibited high polymorphism (100%) (Table 1). This polymorphism was an indication of higher prevalence of diversity among the eight *C. gigantea* genotypes. The results indicated that *C. gigantea* has a considerable number of polymorphisms and genetic variants among its genotypes. As a representative, the amplified diagram produced by the primer UBC-842 is shown in Figure 2.

**Table 1.** Information on ISSR primers among eight populations.

| Primer Name | Primer Sequence (5′-3′) | TNB (*n*) | NPB (*n*) | PPB (%) | Range of the Band Size (bp) |
|---|---|---|---|---|---|
| UBC-808 | $(AG)_8$-C | 6 | 5 | 83.33 | 550~2300 |
| UBC-824 | $(TC)_8$-G | 8 | 8 | 100.00 | 700~3100 |
| UBC-827 | $(AC)_8$-G | 8 | 8 | 100.00 | 750~2400 |
| UBC-836 | $(AG)_8$-YA | 4 | 2 | 50.00 | 500~1500 |
| UBC-841 | $(GA)_8$-YC | 6 | 5 | 83.33 | 200~1200 |
| UBC-842 | $(GA)_8$-YG | 6 | 4 | 66.66 | 200~1800 |
| UBC-847 | $(CA)_8$-RC | 8 | 8 | 100.00 | 800~2400 |
| UBC-856 | $(AC)_8$-YA | 10 | 9 | 90.00 | 500~2400 |
| UBC-857 | $(AC)_8$-YG | 13 | 11 | 84.62 | 300~2600 |
| UBC-873 | $(GACA)_4$ | 9 | 7 | 77.77 | 300~2400 |
| Total | – | 78.00 | 67.00 | – | – |
| Mean | – | 7.80 | 6.70 | 85.90 | – |

Note: TNB, total number of bands; NPB, number of polymorphic bands; PPB (%), percentage of polymorphic bands; and R = (A,G), Y = (C,T).

### 3.2. Genetic Relationship among C. gigantea Genotypes

The observed number of alleles (*Na*) of the eight populations ranged from 1.333 to 1.756, averaging 1.529. The effective number of alleles (*Ne*) spanned from 1.231 (population 4) to 1.489 (population 11), averaging 1.348. The Nei's gene diversity (*H*) was between 0.137 (population 4 and 15) and 0.280 (population 11), with a mean of 0.199. The value of Shannon's information index (*I*) ranged more than two-fold, from 0.199 (population 15) to 0.413 (population 11), with a mean of 0.293. Additionally, the average values of the number of polymorphic loci (*NPLs*) and the percentage of polymorphic loci (*PPLs*) were 41.25 and 52.90, respectively. Among all the populations, population 11 harbored the most genetic diversity (*Ne* = 1.489; *H* = 0.280; *I* = 0.413), whereas population 4 showed the least genetic diversity (*Ne* = 1.231; *H* = 0.137; *I* = 0.205). The order of genetic diversity among the populations in the descending order is as follows: 11 > 13 > 12 > 8 > 14 > 4 > 10 > 15 (Table 2).

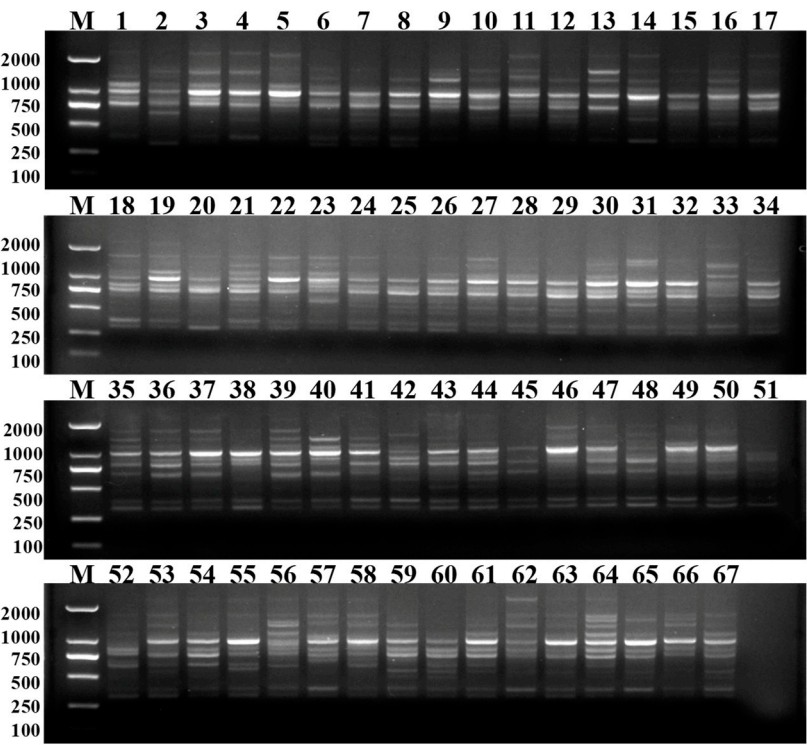

**Figure 2.** Bands produced in all samples using the ISSR marker UBC-842.

**Table 2.** Genetic parameters revealed through ISSR for eight populations of *C. gigantean*.

| Population Name | N | Latitude (N°) | Longitude (E°) | Altitude (m) | Na | Ne | H | I | NPLs | PPLs (%) |
|---|---|---|---|---|---|---|---|---|---|---|
| Population 4 | 4 | 93°26′50.0″ | 29°06′34.5″ | 2989 | 1.385 ± 0.490 | 1.231 ± 0.343 | 0.137 ± 0.188 | 0.205 ± 0.273 | 30 | 38.46 |
| Population 8 | 4 | 93°07′33.8″ | 29°08′00.0″ | 3067 | 1.551 ± 0.501 | 1.381 ± 0.410 | 0.214 ± 0.213 | 0.315 ± 0.302 | 43 | 55.13 |
| Population 10 | 3 | 93°02′25.3″ | 29°03′00.0″ | 3081 | 1.372 ± 0.486 | 1.253 ± 0.369 | 0.145 ± 0.199 | 0.214 ± 0.287 | 29 | 37.18 |
| Population 11 | 17 | 93°14′46.6″ | 28°59′50.0″ | 3105 | 1.756 ± 0.432 | 1.489 ± 0.381 | 0.280 ± 0.196 | 0.413 ± 0.273 | 59 | 75.64 |
| Population 12 | 13 | 93°09′45.0″ | 29°00′50.0″ | 3048 | 1.615 ± 0.490 | 1.387 ± 0.391 | 0.222 ± 0.207 | 0.330 ± 0.293 | 48 | 61.54 |
| Population 13 | 15 | 93°05′06.8″ | 29°05′48.3″ | 3038 | 1.705 ± 0.459 | 1.443 ± 0.384 | 0.255 ± 0.199 | 0.379 ± 0.280 | 55 | 70.51 |
| Population 14 | 7 | 93°09′10.0″ | 29°05′50.0″ | 3012 | 1.513 ± 0.503 | 1.353 ± 0.399 | 0.199 ± 0.213 | 0.292 ± 0.303 | 40 | 50.28 |
| Population 15 | 4 | 93°10′00.0″ | 29°02′50.0″ | 3024 | 1.333 ± 0.475 | 1.250 ± 0.390 | 0.137 ± 0.205 | 0.199 ± 0.291 | 26 | 33.33 |
| Mean | | | | | 1.529 | 1.348 | 0.199 | 0.293 | 41.25 | 52.90 |

N: number of samples per population, *Na*: observed number of alleles, *Ne*: effective number of alleles, *H*: Nei's gene diversity, *I*: Shannon's information index, *NPLs*: the number of polymorphic loci, *PPLs*: the percentage of polymorphic loci.

Using the AMOVA method, the genetic variation among 67 samples from different locations was analyzed (Table 3). Total variation among the population was 14.2%, whereas that within the populations was 85.8%. The results indicated that the substantial genetic diversity of *C. gigantea* mainly came from within the population and genetic variation occurred ($p < 0.001$).

**Table 3.** AMOVA of genetic variance within and among populations based on ISSR data.

| Source | df | MS | Variance Component | Percentage of Variation (%) | Fixation Index |
|---|---|---|---|---|---|
| Among populations | 7 | 22.007 | 1.580 | 14.2 | 0.22480 |
| Within populations | 59 | 9.544 | 9.544 | 85.8 | $p < 0.001$ |
| Total | 66 | 31.551 | 11.124 | 100 | |

*df*, degrees of freedom.

### 3.3. Cluster Analysis in C. gigantea

Binary data obtained for the 10 ISSR primers from 67 samples were scored and computed. Accordingly, a UPGMA tree was built according to Jaccard's coefficient of similarity.

At the threshold value of 0.64, 67 individuals were classified into two categories (Figure 3). Here, two different clusters (Cluster I and II) were derived, in which Cluster I further formed two subclusters labeled here as IA and IB. Subcluster IA contained most of the samples from populations 4, 8, 12, 14, and 15, while subcluster IB included most of the samples from population 13. Cluster II contained three samples from population 10 and one sample from population 11.

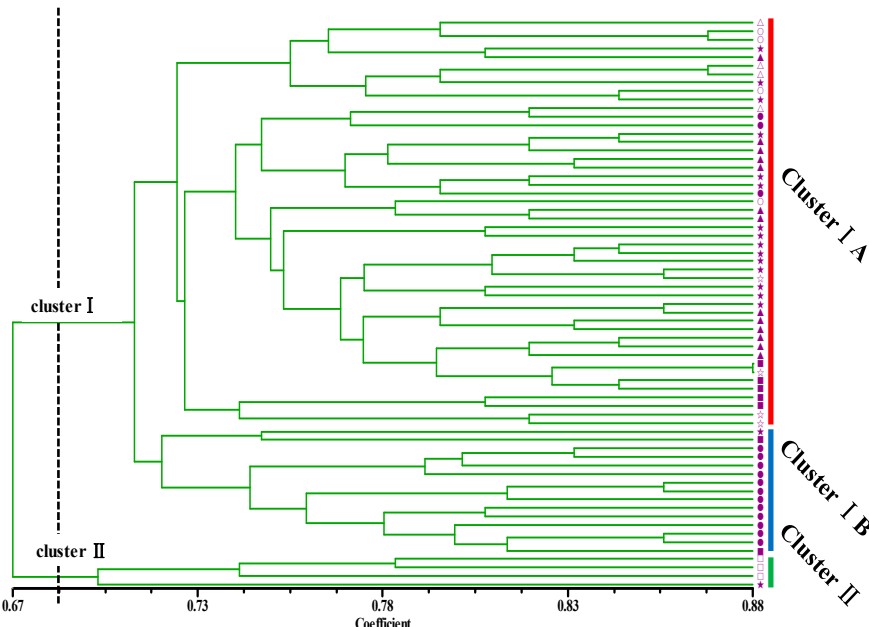

**Figure 3.** UPGMA dendrogram of 67 samples based on Jaccard's coefficient: Δ-samples from population 4; O-samples from population 8; □-samples from population 10; ★-samples from population 11; ▲-samples from population 12; ●-samples from population 13; ■-samples from population 14; and ☆-samples from population 15.

The clustering pattern can be attributed to two factors, including history and environment. Several genotypes stand out as distinct entities within their respective subclusters. Noticeably, individual samples of population 13 in subcluster IA and population 11 in Cluster II exhibit distinct genetic profiles, emphasizing their possible roles in conservation programs.

Using the ISSR data, the dendrogram was generated and the principal coordinate analysis (PCoA) was performed to group the genotypes from the two main clusters in *C. gigantea*. The PCoA also clearly displayed three groupings of the samples, consistent with the above results (Figure 4). The geographical distribution of the population (Figure 1) showed that the habitat conditions divided by longitude and the direction of the river affected the genetic structure of *C. gigantea*.

In order to study the genetic structures of the genotypes of *C. gigantea*, a non-spatial Bayesian clustering method was used to determine the optimal number of subpopulations (K) (Figure 5). These results demonstrated that the fitted model with $K = 3$ robustly explained the data. Red, green, and blue vertical bars represented the three main groups. This result was consistent with that obtained from the UPGMA cluster analysis as well as that from the PCoA analysis (Figures 3 and 4). These results indicate that the individuals of the *C. gigantea* population may be genetically derived from three gene pools.

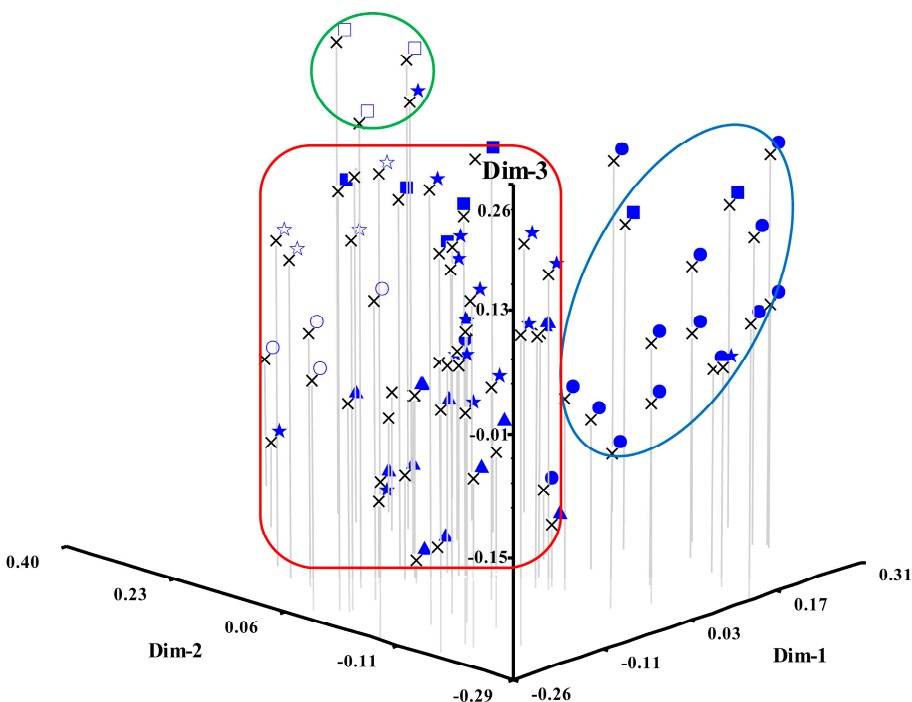

**Figure 4.** Principal coordinate analysis for the ISSR data: Δ-samples from population 4; O-samples from population 8; □-samples from population 10; ★-samples from population 11; ▲-samples from population 12; ●-samples from population 13; ■-samples from population 14; and ☆-samples from population 15; red border represents Cluster IA; blue border represents Cluster IB; green border represents Cluster II.

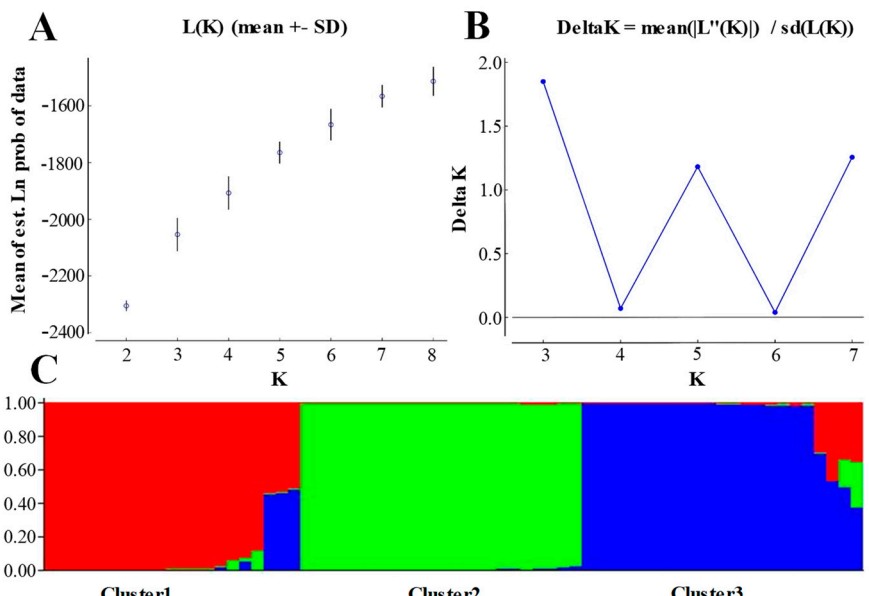

**Figure 5.** Population structure for $K = 3$ based on the ISSR data: (**A**)-mean of likelihood value L (K) of each K; (**B**)-optimum group numbers of tested apricot samples inferred by ΔK; and (**C**)-genetic structure of 67 individuals analyzed with STRUCTURE ($K = 3$). Red, green, and blue color vertical bars represent the genotypes from different populations.

## 4. Discussion

The identification of genetic diversity is important for understanding plant evolution and adaptation [43]. In general, those plant species with genetic variability have the potential to adapt to extreme and unique environmental conditions [44]. Consequently, it is

important to assess the genetic diversity levels of tree species for tree management and conservation strategies, especially for rare tree species [26]. Ample evidence on the pattern and distribution of genetic variation in conifers indicates that they have higher levels of genetic variation than other plant species, with little differentiation among populations [45,46].

Herein, a high level of genetic diversity occurred within populations, which is consistent with a previous report [12]. Our results revealed that ISSR markers were efficient molecular markers for evaluating the genetic polymorphism among different *C. gigantea* populations because three out of the ten ISSR markers that we used displayed 100% polymorphism (Table 1). Noteworthily, a high level of among-population genetic diversity was also reported with AFLP markers in *C. gigantea* [26]. Similar results were found in other research studies on conifer trees. For example, both the RAPD and ISSR methods indicated that rich genetic diversity and most variation came from within populations in Larix gmelinii [47]. Subsequently, by using six RAPD and three ISSR markers for four subpopulations of Abies cephalonica, Papageorgiou et al. demonstrated that all subpopulations exhibit high genetic diversity, with relatively low levels of differentiation among them [48]. More recently, another group discovered that the 129 individuals revealed a high level of genetic diversity and 85% of genetic variation within populations in Tetraclinis articulate [49]. This may be because *C. gigantea* has an outcrossing breeding system with high potential for long-distance gene flow.

Therefore, ISSR markers are highly sensitive and reproducible methods to detect the population structure of *C. gigantea*. For other plant species, ISSR primers were found to be more effective than other types of molecular markers [50,51]. For instance, Linos et al. proposed that the average number of alleles per locus was 5.8 for SSR, while ISSR and RAPD showed lower values (2 and 2, respectively), indicating somewhat lower congruence with the dominant marker data [52]. Another group also demonstrated that the polymorphism of ten ISSR primers (100%) is higher than SSR (80.4%), and significant differences were found in three distinct groups among the genotypes in chrysanthemum [53]. However, combining and contrasting the findings of the ISSR technique with those derived from other molecular methodologies, such as AFLP, SNP, or next-generation sequencing, can enrich the understanding of the genetic diversity and configuration of *C. gigantea* populations [54]. In summary, when assessing large numbers of samples, it is important to combine multiple and reliable methods to evaluate the genetic diversity of different plant species.

The genetic diversity differed greatly between all studied populations (Table 2). Accordingly, this result could be helpful to analyze the current status of different *C. gigantea* populations, leading to sound conservation strategies. For example, because populations 4, 10, and 15 had lower genetic diversity (Table 2), they should be declared as critical populations, as they are at risk of extinction. Habitat fragmentation reduces the gene flow among populations, resulting in a loss of genetic diversity. Therefore, conservation strategies should be immediately implemented such as ex situ collections (such as the collection of germplasm and the establishment of seed banks) and the introduction of new germplasm and the establishment of breeding systems (including cutting, grafting propagation, and tissue culture). In addition, populations 11 and 13 exhibited high levels of genetic diversity (Table 2). These populations represent the core populations with strong environmental adaptability and evolutionary potential. Hence, regulations and management strategies must be established to protect the natural habitat of *C. gigantea*. Notably, national-level core germplasm banks of *C. gigantea* should be established to maintain the genetic variation and develop new genetic breeding materials.

Furthermore, the results of AMOVA found that the average genetic variation within population was higher than among populations, indicating high gene flow between populations (Table 3). This result has been attributed primarily to the long-distance dispersal of pollen and seeds, the longevity, and the outcrossing of *C. gigantean* with other conifers. In general, the establishment of in situ protected areas is an ideal way to protect wild plants. However, the complex terrain and diverse climate of the Tibetan plateau make it unsuitable for the conservation of endangered plant species. Strikingly, in situ conservation also re-

quires local authorities to establish appropriate policies [55]. Correspondingly, considering these factors, an ex situ conversation based on a seed bank is a feasible option to protect the germplasm resources of *C. gigantean*. At the same time, promoting the cultivation and domestication of this wild plant is of great significance for meeting the market demand and protecting endangered plant species.

Some factors may contribute to the reduction in the high genetic diversity of *C. gigantea*, including natural selection and human impact. The first factor can be the response of Tibet's Yarlung Tsangbo River to global climate change; conversely, the generation of complex landscapes under climate fluctuations may partly account for the high genetic diversity of *C. gigantea* [56,57]. The second factor can be logging and other forms of human damage to trees [58]. Here, the AMOVA results showed a higher magnitude of genetic differentiation within the populations of *C. gigantea*. This result probably reflects the frequent gene flow occurring among *C. gigantea* populations because seeds and pollen of *C. gigantea* can be dispersed to long distances. Therefore, marked geographic barriers of and climate changes in the Yarlung Tsangbo River are not factors limiting the gene flow of *C. gigantea*. Furthermore, the suitable habitat of *C. gigantea* faces various threats; therefore, an assessment of *C. gigantea*'s in situ genetic diversity would be beneficial for its development and genetic protection in future. The molecular marker technique is an efficient tool for distinguishing *C. gigantea* populations and for robustly identifying the populations most at risk.

## 5. Conclusions

Herein, we used ISSR to analyze the genetic structure of *C. gigantea* populations collected along the north of the deep gorge of the Yarlung Tsangbo River. The results show that a large proportion of genetic diversity is present at the within-population level in *C. gigantea*. Additionally, UPGMA clustering together with PCoA analysis revealed that all accessions can be divided into two main groups. Furthermore, marked geographic barriers of and climate changes in the Yarlung Tsangbo River were not factors limiting gene flow of *C. gigantea*. Accordingly, a gene bank of *C. gigantea* should be founded by collecting more germplasm around the Yarlung Tsangbo River.

**Author Contributions:** F.M., P.L., X.J. and Y.J. conducted the experiments. X.J., Y.J. and J.L. wrote the manuscript. F.M., X.J., Y.J. and J.L. edited the manuscript. F.M., P.L. and X.J. critically revised the draft and updated the manuscript for publication. All authors have read and agreed to the published version of the manuscript.

**Funding:** This study was supported by the Fundamental Research Funds for the Central Universities (2572018CG05).

**Data Availability Statement:** Data are contained within the article.

**Acknowledgments:** This study was supported by the Fundamental Research Funds for the Central Universities (2572018CG05).

**Conflicts of Interest:** The authors declare no conflicts of interest.

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
