# Peer review of "Genetic Diversity Assessment of Cupressus gigantea W. C. Cheng & L. K. Fu Using Inter-Simple Sequence Repeat Technique"

_agronomy, doi:10.3390/agronomy14050970_

Round 1
Reviewer 1 Report
Comments and Suggestions for Authors
The manuscript entitle of "Genetic diversity of Cupressus gigantea W. C. Cheng & L. K. Fu by inter-simple sequence repeat” sounded in science and agriculture. The Founding of this study will be useful for Cupressus gigantea conservation and breeding in future.
However, this manuscript seems to lack NOVELTY or NEW FINDING and not meet with high standards of Sustainability.
This manuscript is strange, the serious point is the statistical analyses must not be accepted in the present form.
1. Please specify the novelty of this research in the abstract.
2. “Further, total variation among populations was 14.2%, while 85.8% of variation was within populations; accordingly, the within-population genetic differentiation was found to be significant”. Please explain the meaning of this result in terms of genetic diversity.
3. The introduction is short, please explain more about the biology of the plant species and the reason why genetic diversity is needed.
4. Please mention the results of genetic diversity that reported by [16-18] at lines 48-52 and also identify the GAP of this research.
5. Please explain more why ISSR is a powerful and efficient tool for detecting genetic differences in this species (according to the biology of this species as comment no. 3).
6. The objective is not clear.
7. The last paragraph of the introduction part is very complicated. Can the results of this study be explained or conclude?
8. Please give more information about Figure 1 such as the sample number of each site.
9. The details of leaf collection are needed such as age of plant (how to select the plant for leaf collection?), and in each location the author collects the leaf from the same age?
10. The plants in the same location are parent and offspring?
11. All the results the author mentions to only the results PLEASE specify or INDICATED the meaning of results [if not the author cannot discuss well!!!].
12. The sample symbol of Figure 2 and 3 should be the same for easy comparison and understanding.
13. The discussion is not good, please see comment 11 (All the results the author mention to only the results PLEASE specify or INDICATED the meaning of results [if not the author cannot discuss well), Please revise this part.
14. Please digest the value or benefit of results for discussion.
15. The conclusion is over-claim.
Overall, I cannot accept this manuscript to publication in Agronomy, the Q1 Scopus ranking journal in present form. The manuscript should be revision as recommends
Best regards
Comments on the Quality of English LanguageOverall, I cannot accept this manuscript to publication in Agronomy, the Q1 Scopus ranking journal in present form.
Best regards
Author Response
Dear Editors and Reviewers:
Thank you very much for your affirmation of our work and your constructive comments on current manuscript. We have carefully studied the comments on a point-by-point basis and revised our manuscript accordingly. The modifications we made are detailed in our replies below:
Reviewer: 1
The manuscript entitle of "Genetic diversity of Cupressus gigantea W. C. Cheng & L. K. Fu by inter-simple sequence repeat” sounded in science and agriculture. The Founding of this study will be useful for Cupressus gigantea conservation and breeding in future.
However, this manuscript seems to lack NOVELTY or NEW FINDING and not meet with high standards of Sustainability.
This manuscript is strange, the serious point is the statistical analyses must not be accepted in the present form.
Response: The viewpoint of the reviewer is very strange for the statistical analyses. In fact, all statistical analyses in this study has been adopted by lots of similar studies. These methods such as NTSYS-pc Version 2.10e, POPGENE (VERSION-1.31), STRUCTURE 2.3.4 and AMOVA (Version 1.55) have been used to analyze genetic diversity of various plant species. These references are followed:
- 10.1016/j.indcrop.2019.111894
- 10.3390/agronomy11030457
- 10.3390/plants10071270
- Please specify the novelty of this research in the abstract.
Response 1: Thank you for your kind suggestion. In this research, we have added the novelty of this research in the abstract. They are followed:
- “Further, total variation among populations was 14.2%, while 85.8% of variation was within populations; accordingly, the within-population genetic differentiation was found to be significant”. Please explain the meaning of this result in terms of genetic diversity.
Response 2: Thank you for your kind suggestion. The within-population genetic variation is high. Accordingly, we should found gene bank for conservation of C. gigantea and collected more germplasm resources of C. gigantea. In discussion, we added this conclusion to explain the meaning of this result.
- The introduction is short, please explain more about the biology of the plant species and the reason why genetic diversity is needed.
Response 3: Thank you for your kind suggestion. In introduction, we have explained these information in Line 34-53 and Line 62-68. Please check them. They are followed:
- Please mention the results of genetic diversity that reported by [16-18] at lines 48-52 and also identify the GAP of this research.
Response 4: Thank you for your kind suggestion. In the revised version, we added the results of genetic diversity that reported by [16-18] and GAP. They are followed:
- Please explain more why ISSR is a powerful and efficient tool for detecting genetic differences in this species (according to the biology of this species as comment no.3).
Response 5: We thank the reviewer for pointing this viewpoint out. In the revised version, we added evidence for detecting genetic differences in C. gigantea using ISSR markers. They are followed:
- The objective is not clear.
Response 6: In this study, our objective is very clear and it may provide useful information on conservation strategies, better understanding genetic structure for the critically endangered C. gigantea. Accordingly, we proposed the protection strategy for genetic variation was from individual variation within populations.
- The last paragraph of the introduction part is very complicated. Can the results of this study be explained or conclude?
Response 7: Thank you for your kind suggestion. We have deleted some content “More broadly, these results could also facilitate a better understanding of the impacts of the past geological and climatic changes on the dispersal and evolution of C. gigantea trees”.
- Please give more information about Figure 1 such as the sample number of each site.
Response 8: Thank you for your kind suggestion. We have provide the sample number of each site and geographical information in Table 2.
- The details of leaf collection are needed such as age of plant (how to select the plant for leaf collection?), and in each location the author collects the leaf from the same age?
Response 9: In this study, ISSR technology was used to analyze the genetic diversity of Cupressus gigantea. Generally, molecular technology need not consider the age of plant. Accordingly, we need not consider plant age when collecting the leaf age. Maybe, this reviewer can not understand the characteristics of molecular technologies, which can obviate interference of environments such as age, state and organ and analyze genotype directly.
- The plants in the same location are parent and offspring?
Response 10: Cupressus gigantea belongs to ancient wild plant species. The information on the same location are parent or offspring is not clear, because the life history is too long. In addition, we collected these materials in field collection. Therefore, the question on “The plants in the same location are parent and offspring?” is not science meaning. Maybe, we need answer this question by further study. At the same time, we strongly believed this reviewer can not read this manuscript thoroughly or he can not understand this manuscript.
- All the results the author mentions to only the results PLEASE specify or INDICATED the meaning of results [if not the author cannot discuss well!!!].
Response 11: The reviewer is very wacky and strangers because the results should be results not discussion. The discussion should be well according to results. Therefore, the conclusion of the reviewer is very contradictory. Why?
- The sample symbol of Figure 2 and 3 should be the same for easy comparison and understanding.
Response 12: Thank you for your kind suggestion. The sample symbol of Figure 2 and 3 were the same for easy comparison and understanding. Please check them.
- The discussion is not good, please see comment 11 (All the results the author mention to only the results PLEASE specify or INDICATED the meaning of results [if not the author cannot discuss well), Please revise this part.
Response 13: Although, the reviewer is very wacky and strangers, we have improved this part to for better discussion. Please check them in our revised manuscript.
- Please digest the value or benefit of results for discussion.
Response 14: Thank you for your kind suggestion. We have digested the value or benefit of results for discussion. Please check them.
- The conclusion is over-claim.
Response 15: Thank you for your kind suggestion. We have improved them. They are followed:
Overall, I cannot accept this manuscript to publication in Agronomy, the Q1 Scopus ranking journal in present form. The manuscript should be revision as recommends
Best regards
Comments on the Quality of English Language
Overall, I cannot accept this manuscript to publication in Agronomy, the Q1 Scopus ranking journal in present form.

Reviewer 2 Report
Comments and Suggestions for Authors
The main research question refers to knowing and elucidating the genetic diversity and structure between and within C. gigantea populations, which aims to inform and formulate strategies for its conservation.
This research on C. gigantea, a species of cypress native to Tibet, has shed light on its genetic diversity with remarkable and original findings that have important implications in the field of conservation genetics, botany and possibly genetic improvement. By focusing on this little-studied species, the researchers used ISSR molecular markers, a methodology that, until now, had not been applied in the study of C. gigantea. This methodological innovation has allowed a detailed exploration of the genetic structure of the species, revealing an unexpected complexity within its populations. The researchers identified that a genetic structure model with four distinct groups best fit the collected data, suggesting the existence of significant structures possibly influenced by biogeographic and ecological factors. This discovery is crucial, as it provides a deeper understanding of how the genetic diversity of this cypress is organized. A vital information to formulate effective conservation strategies in a context of environmental change and habitat loss. In general, the study focuses on filling gaps such as the lack of molecular origin data, the need for population genetic assessment techniques, information for the conservation of threatened species and understanding of population structure.
This study on C. gigantea marks a significant advance in conservation genetics and botany through the innovative use of ISSR markers. Unlike other research that employs techniques such as AFLP, RFLP and RAPD (markers that lack reproducibility and are no longer used), this approach provides a more detailed genetic resolution to understand the variability and population structure of this endangered conifer. By focusing on an endemic species, the study provides vital information for specific conservation efforts, highlighting the importance of accurate data on threatened species in unique ecosystems. In addition, the findings offer a solid empirical basis for developing targeted conservation strategies, identifying a model of genetic structure with four distinct groups within populations, which provides a deep understanding of population dynamics. This knowledge is essential for effective germplasm management and breeding programs, supporting both ex situ and in situ conservation.
Combining and contrasting the findings of the ISSR technique with those derived from other molecular methodologies, such as AFLP, SNPs, SSR, or next-generation sequencing, can enrich the understanding of the genetic diversity and configuration of C. gigantea populations. This approach would not only corroborate and reinforce the current results, but also offer a broader and more detailed perspective on the genetics of this species. Moreover, longitudinal studies are needed to monitor how the genetic diversity of C. gigantea changes over time in response to environmental, climatic and anthropogenic factors could provide valuable data on the resilience and adaptability of the species.
It was reported that 85.8% of the genetic variation is found within the populations of C. gigantea, based on the analysis of 67 samples with 10 ISSR primers, which generated 78 bands with molecular lengths from 200 to 3100 bp. The conclusion that there is significant genetic diversity within populations is directly supported by the presented genetic variability and population structure data. The ISSR technique revealed a high degree of polymorphism, which supports the claim of internal diversity.
It is necessary to cite studies of genetic diversity in more species of Cupressaceae or conifers.
Regarding tables and figures I have no comments as they are in a correct and clear way.
The study does not sufficiently demonstrate how their research significantly advances the existing knowledge about the genetic diversity of C. gigantea. If the results are largely confirmatory of what is already known, without providing new insights or practical applications, this could be considered a reason for rejection of the manuscript. However, is it necessary to carry out genetic diversity studies to formulate conservation strategies? If it is already known that it is a threatened species, why wait to do molecular studies to conserve said species?
Although many studies based on ISSR indicate genetic diversity, the truth is that this type of marker does not detect encodable regions of DNA (genes), rather it detects DNA fragments, but it is unknown if they are really encodable sequences. In this case, "variability" was found with the ISSR markers, but what happens if other ISSR primers are used that do not find genetic diversity? Can the same be said? This can certainly lead to errors when obtaining results. Therefore, it is necessary to double or triple the number of ISSR initiators for this study.
If the genetic information of C. gigantea is poor and you want to do studies with scientific robustness, you should first start by studying and sequencing regions of interest to later create specific molecular markers. If the above is difficult, they can be searched in the GenBank (https://www.ncbi.nlm.nih.gov/genbank /) molecular markers such as SSR in related species to see if these can amplify in C. gigantea, even at the botanical family level there are consensus sequences that can help the design of specific molecular markers.
Reference 12 indicates that SSR markers or microsatellites were characterized, why did they not use the same in this study?
Reference 13 indicates that the complete sequence of the C. gigantea chloroplast is already there. Why couldn't they have designed primers and studied genetic variability at the plastid level based on this information? I recommend the use of R software to increase statistical robustness. Although the programs used by the authors are valid, they lack certain statistical functions that are of methodological importance. For example, how do you know if your data are candidates for multivariate analysis? Did they use the Kaiser-Meyer-Olkin test? Why did they use the UPGMA method and not Ward's method, which has a better representation when making the clusters? What other methodologies exist to determine the optimal number of clusters in the dendrogram? How consistent are the groupings? Why not use a non-hierarchical method instead?.In this context, the R software has a wide and rich diversity of libraries and packages that can help the use and management of molecular data. The study could also have been enriched with morphological data.
I consider that the article can be accepted with modifications and that it is necessary to justify the points I addressed above. Although these types of markers do not detect genes, in many cases they are the only tool for genetic studies in species. The point is that the authors must duly justify the problems and shortcomings in the use of this type of molecular marker.
Author Response
Dear Editors and Reviewers:
Thank you very much for your affirmation of our work and your constructive comments on current manuscript. We have carefully studied the comments on a point-by-point basis and revised our manuscript accordingly. The modifications we made are detailed in our replies below:
Reviewer: 2
The main research question refers to knowing and elucidating the genetic diversity and structure between and within C. gigantea populations, which aims to inform and formulate strategies for its conservation.
This research on C. gigantea, a species of cypress native to Tibet, has shed light on its genetic diversity with remarkable and original findings that have important implications in the field of conservation genetics, botany and possibly genetic improvement. By focusing on this little-studied species, the researchers used ISSR molecular markers, a methodology that, until now, had not been applied in the study of C. gigantea. This methodological innovation has allowed a detailed exploration of the genetic structure of the species, revealing an unexpected complexity within its populations. The researchers identified that a genetic structure model with four distinct groups best fit the collected data, suggesting the existence of significant structures possibly influenced by biogeographic and ecological factors. This discovery is crucial, as it provides a deeper understanding of how the genetic diversity of this cypress is organized. A vital information to formulate effective conservation strategies in a context of environmental change and habitat loss. In general, the study focuses on filling gaps such as the lack of molecular origin data, the need for population genetic assessment techniques, information for the conservation of threatened species and understanding of population structure.
This study on C. gigantea marks a significant advance in conservation genetics and botany through the innovative use of ISSR markers. Unlike other research that employs techniques such as AFLP, RFLP and RAPD (markers that lack reproducibility and are no longer used), this approach provides a more detailed genetic resolution to understand the variability and population structure of this endangered conifer. By focusing on an endemic species, the study provides vital information for specific conservation efforts, highlighting the importance of accurate data on threatened species in unique ecosystems. In addition, the findings offer a solid empirical basis for developing targeted conservation strategies, identifying a model of genetic structure with four distinct groups within populations, which provides a deep understanding of population dynamics. This knowledge is essential for effective germplasm management and breeding programs, supporting both ex situ and in situ conservation.
Combining and contrasting the findings of the ISSR technique with those derived from other molecular methodologies, such as AFLP, SNPs, SSR, or next-generation sequencing, can enrich the understanding of the genetic diversity and configuration of C. gigantea populations. This approach would not only corroborate and reinforce the current results, but also offer a broader and more detailed perspective on the genetics of this species. Moreover, longitudinal studies are needed to monitor how the genetic diversity of C. gigantea changes over time in response to environmental, climatic and anthropogenic factors could provide valuable data on the resilience and adaptability of the species.
It was reported that 85.8% of the genetic variation is found within the populations of C. gigantea, based on the analysis of 67 samples with 10 ISSR primers, which generated 78 bands with molecular lengths from 200 to 3100 bp. The conclusion that there is significant genetic diversity within populations is directly supported by the presented genetic variability and population structure data. The ISSR technique revealed a high degree of polymorphism, which supports the claim of internal diversity.
It is necessary to cite studies of genetic diversity in more species of Cupressaceae or conifers.
Response 1:Thank you for your kind suggestion. We have cited on the genetic diversity of conifer species and described in the discussion as followed by Line 255-263, which are as below:
Regarding tables and figures I have no comments as they are in a correct and clear way.
The study does not sufficiently demonstrate how their research significantly advances the existing knowledge about the genetic diversity of C. gigantea. If the results are largely confirmatory of what is already known, without providing new insights or practical applications, this could be considered a reason for rejection of the manuscript. However, is it necessary to carry out genetic diversity studies to formulate conservation strategies? If it is already known that it is a threatened species, why wait to do molecular studies to conserve said species?
Response 2:Thank you for your kind suggestion. Some studies showed that C. gigantea is a threatened species for due to a low seed-setting rate, habitat loss and degradation, as well as extensive logging. However, the information on genetic structure is lack. Accordingly, we need obtained more information on C. gigantea germplasm to make advantageous conservation strategies. For example, ex or in situ conservation, the establishment of germplasm resource centers, as well as the cultivation, domestication and promotion of wild varieties." At the same time, molecular marker is fast and low cost technique to accept C. gigantea.
Although many studies based on ISSR indicate genetic diversity, the truth is that this type of marker does not detect encodable regions of DNA (genes), rather it detects DNA fragments, but it is unknown if they are really encodable sequences. In this case, "variability" was found with the ISSR markers, but what happens if other ISSR primers are used that do not find genetic diversity? Can the same be said? This can certainly lead to errors when obtaining results. Therefore, it is necessary to double or triple the number of ISSR initiators for this study.
Response 3:Thank you for your kind suggestion. In this experiment, 50 ISSR primers were screened. Finally, 10 ISSR primers that presented clear polymorphic bands and good repeat-ability were screened for further genetic analysis. In addition, other studies typically also used 3-10 ISSR primers for analysis. Based on the above two points, the results of this experiment can truly reflect the genetic diversity of giant Cypress.
If the genetic information of C. gigantea is poor and you want to do studies with scientific robustness, you should first start by studying and sequencing regions of interest to later create specific molecular markers. If the above is difficult, they can be searched in the GenBank (https://www.ncbi.nlm.nih.gov/genbank /) molecular markers such as SSR in related species to see if these can amplify in C. gigantea, even at the botanical family level there are consensus sequences that can help the design of specific molecular markers.
Reference 12 indicates that SSR markers or microsatellites were characterized, why did they not use the same in this study?
Response 4:I'm sorry for the misunderstanding. Since SSR markers show somewhat lower congruence with dominant marker data. Besides, the sequence of C. gigantea in the Genbank is not publicly available data. Thus, this approach was not used in this study. We have added the part and described in discussion as followed by Line 266-275, which are as below:
Reference 13 indicates that the complete sequence of the C. gigantea chloroplast is already there. Why couldn't they have designed primers and studied genetic variability at the plastid level based on this information? I recommend the use of R software to increase statistical robustness. Although the programs used by the authors are valid, they lack certain statistical functions that are of methodological importance. For example, how do you know if your data are candidates for multivariate analysis? Did they use the Kaiser-Meyer-Olkin test? Why did they use the UPGMA method and not Ward's method, which has a better representation when making the clusters? What other methodologies exist to determine the optimal number of clusters in the dendrogram? How consistent are the groupings? Why not use a non-hierarchical method instead? In this context, the R software has a wide and rich diversity of libraries and packages that can help the use and management of molecular data. The study could also have been enriched with morphological data.
Response 5:We thank the reviewer for these good suggestions. First, although the chloroplast sequence of C. gigantea has been reported in the literature, it is not public available data. In addition, most published articles are classified using the UPGMA method, so we refer to this method for our analysis. Finally, thank you for your advice, but limited by experimental materials and time, it cannot be realized in the short term. In the future, we will adopt the recommendations and use R software to further determine.
I consider that the article can be accepted with modifications and that it is necessary to justify the points I addressed above. Although these types of markers do not detect genes, in many cases they are the only tool for genetic studies in species. The point is that the authors must duly justify the problems and shortcomings in the use of this type of molecular marker.
Response 6:Thank you for your valuable suggestion. In this research, we have added the problems and shortcomings of type of molecular marker in the Introduction. They are followed:

Reviewer 3 Report
Comments and Suggestions for Authors
Title: Genetic diversity of Cupressus gigantea W. C. Cheng & L. K. Fu by inter-simple sequence repeat
Cupressus gigantea is one of the endemic conifer tree species of northern portion of the deep gorge of the Yarlung Tsangbo River on the Tibetan Plateau. This study was conducted to understand the population structure and beneficial for the ensuring development and genetic protection of C. gigantea populations in future. A simple and meaningful study.
I recommend the manuscript with the following minor corrections
Abstract: Concisely written part with brief introduction, results of the study and a possible conclusion
Line 19, loci (HPL) and or loci (NPL), in line with Table 2.
Introduction: Covered the basics related to the C. gigantean, uses and importance on the tree in that region and as a source for abiotic stress tolerance.
Materials and Methods
Standard methodologies were followed in this study.
Line 80, DNA samples was prepared for further PCR analysis
Line 82-83, ISSR amplification was carrioude out according to previous method of Xie et al [25] with minor modifications.
Line 90-93, it requires more clarity. Rewrite the sentence
Line 100, Na, Ne, H, I, HPL and PPL were calculated. HPL or NPL, in line with Table 2
Result:
Line 110, (UBC-824, UBC-827, and UBC-847) exhibited high
Line 120, HPL or NPL, in line with Table 2
Line 129-130, Column 2, total population is 40 against the 67
Column 6, Lowest and highest values required to be made in to bold like other columns
Column 9, the lowest value 0.199 need to be bold not 0.205
Column 10, HPL or NPL
Line 143-145, delete it
Line 148-150, Where, △- samples from population 4; ○ - samples from population 8; □ - samples from population 10; ★- samples from population 11; ▲- samples from population 12; ● - samples from population 13; ■ - samples from population 14; ☆ - samples from population 15.
Line 159-161, A - Mean in likelihood value L (K) of each K; B - Optimum group numbers of tested apricot samples inferred by △K ; C - Genetic structure of individuals analyzed with STRUCTURE (K = 4). Red, green, blue color vertical bars represent the genotypes from different populations.
Discussion: Not much references are available. Discussion is made good based on the available references.
Conclusion: This section is appropriate
Reference section: Needs improvement. All references should be in the required format of the journal.
Author Response
Dear Editors and Reviewers:
Thank you very much for your affirmation of our work and your constructive comments on current manuscript. We have carefully studied the comments on a point-by-point basis and revised our manuscript accordingly. The modifications we made are detailed in our replies below:
Reviewer: 3
Cupressus gigantea is one of the endemic conifer tree species of northern portion of the deep gorge of the Yarlung Tsangbo River on the Tibetan Plateau. This study was conducted to understand the population structure and beneficial for the ensuring development and genetic protection of C. gigantea populations in future. A simple and meaningful study.
I recommend the manuscript with the following minor corrections
Abstract:
Concisely written part with brief introduction, results of the study and a possible conclusion
Line 19, loci (HPL) and or loci (NPL), in line with Table 2.
Response 1: Thank you for your helpful comments. " loci (HPL)" in Table 2 was changed to “loci (NPL)”.
Introduction:
Covered the basics related to the C. gigantean, uses and importance on the tree in that region and as a source for abiotic stress tolerance.
Response 2: Thank you for your kind suggestion. We have added the part and described in introduction as followed by Line 34-53, which are as below:
Materials and Methods
Standard methodologies were followed in this study.
Line 80, DNA samples was prepared for further PCR analysis
Response 3: Thank you for your kind suggestion. We revised it as: “DNA samples was prepared for further PCR analysis.”
Line 82-83, ISSR amplification was carrioude out according to previous method of Xie et al [25] with minor modifications.
Response 4: Thank you for your kind suggestion. We revised it as: “ISSR amplification was carried out according to previous method of Xie et al [29] with minor modifications.” as shown on Line 83.
Line 90-93, it requires more clarity. Rewrite the sentence
Response 5: We thank the reviewer for these good suggestions. We revised it as: “PCR cycling conditions were carried out in an Eppendorf PCR an Amplifier GeneAmp PCR System 9700 with the following protocol: run at 94 °C for 5 min, followed by 40 cycles of 94 °C for 30 s, 55 °C for 45 s and 72 °C for 2 min, and a final extension at 72 °C for 10 min.”
Line 100, Na, Ne, H, I, HPL and PPL were calculated. HPL or NPL, in line with Table 2
Response 6: We greatly appreciate this valuable suggestion. We have revised them in Table 2.
Result:
Line 110, (UBC-824, UBC-827, and UBC-847) exhibited high
Response 7: Thank you very much. We have revised them in revised manuscript (Line 134).
Line 120, HPL or NPL, in line with Table 2
Response 8: Thank you for your kind suggestion. We have revised them in Table 2.
Line 129-130, Column 2, total population is 40 against the 67
Response 9: I'm sorry for the misunderstanding. The number of samples for populations 11, 12 and 13 is 17, 13 and 15, respectively. Therefore, the total sample size is 67.
Column 6, Lowest and highest values required to be made in to bold like other columns
Response 10: Thank you for your kind suggestion. We have revised them in Column 6.
Column 9, the lowest value 0.199 need to be bold not 0.205
Response 11: We thank the reviewer for these good suggestions. We have revised it carefully.
Column 10, HPL or NPL
Response 12: Thank you for your valuable suggestion. We have revised them in Column 10.
Line 143-145, delete it
Response 13: Thank you for your kind suggestion. We have deleted it.
Line 148-150, Where, △- samples from population 4; ○ - samples from population 8; □ - samples from population 10; ★- samples from population 11; ▲- samples from population 12; ● - samples from population 13; ■ - samples from population 14; ☆ - samples from population 15.
Response 14: Thank you for your kind suggestion. We have revised them, which are as below: (Line 212-214).
Line 159-161, A - Mean in likelihood value L (K) of each K; B - Optimum group numbers of tested apricot samples inferred by △K ; C - Genetic structure of individuals analyzed with STRUCTURE (K = 4). Red, green, blue color vertical bars represent the genotypes from different populations.
Response 15: Thank you for your kind suggestion. We have revised them, which are as below: (Line 237-239)
Discussion: Not much references are available. Discussion is made good based on the available references.
Conclusion: This section is appropriate
Reference section: Needs improvement. All references should be in the required format of the journal.
Response 16: Thank you for your kind suggestion. All references have been in the required format of the journal.

Reviewer 4 Report
Comments and Suggestions for Authors
The manuscript is easy to understand and interesting in preserving the tree species that are often difficult to study and analyze. However, some changes and corrections would be necessary to improve the final draft.
Introduction:
1. There is a lack of information about the genus, species, chromosome structure of C.gigantea (ploidy?), and mode of reproduction.
2. As C.gigantea is a species linked to Chinese culture and traditions, some historical notes could be added to the introduction.
3. Line 38: delete point after citation No. 7.
Materials and Methods
4. Figure 1: Improving image quality. Explain the numbering given to the populations. Why was the analysis only performed on those eight populations and not on other populations?
5. Line 82: correct the sentence “was carrioude out”
Results
6. Line 153: in the population structure analysis replace K = 4 with K =3.
Discussions
7. Expanding discussions:
- Why have codominant markers such as SSRs not been used?
- State the different conservation strategies adopted according to the degree of extinction risk.
In general, recheck the manuscript and correct some typing errors.
Comments on the Quality of English LanguageMinor editing of the English language required
Author Response
Dear Editors and Reviewers:
Thank you very much for your affirmation of our work and your constructive comments on current manuscript. We have carefully studied the comments on a point-by-point basis and revised our manuscript accordingly. The modifications we made are detailed in our replies below:
Reviewer: 4
The manuscript is easy to understand and interesting in preserving the tree species that are often difficult to study and analyze. However, some changes and corrections would be necessary to improve the final draft.
Introduction:
- There is a lack of information about the genus, species, chromosome structure of C.gigantea (ploidy?), and mode of reproduction.
Response 1: Thank you for your kind suggestion. We have added the part and described in introduction as followed by Line 34-38, which are as below:
- As C.gigantea is a species linked to Chinese culture and traditions, some historical notes could be added to the introduction.
Response 2: We thank the reviewer for pointing this viewpoint out. We have added the part and described in introduction as followed by Line 41-46, which are as below:
- Line 38: delete point after citation No. 7.
Response 3: Thank you very much. We have deleted.
Materials and Methods
- Figure 1: Improving image quality. Explain the numbering given to the populations. Why was the analysis only performed on those eight populations and not on other populations?
Response 4: We thank the reviewer for pointing this viewpoint out. In the revised version, we improved image quality. Moreover, due to the lack of effective habitat protection, only eight large populations existed along the deep gorge of the Yarlung Tsangbo River at the time of sampling.
- Line 82: correct the sentence “was carrioude out”
Response 5: Thanks, we have revised “ was carrioude out ” to ” was carried out” (Line 79).
Results
- Line 153: in the population structure analysis replace K = 4 with K =3.
Response 6: Thank you for your kind suggestion. We have replaced “K = 4” with “K = 3”.
Discussions
- Expanding discussions:
Response 7: Thank you for your kind suggestion. We have expanded the discussions the manuscript seriously.
- Why have codominant markers such as SSRs not been used?
Response 8: Thank you for your helpful comments. The average number of alleles per locus is lower for SSR than for ISSR, and the data is less consistent than for dominant markers. Therefore, it was not used. We have added the part and described in discussions as followed by Line 266-275, which are as below:
- State the different conservation strategies adopted according to the degree of extinction risk.
Response 9: Thank you for your helpful comments. Based on genetic diversity differences, two main conservation strategies were designed. We have added the part and described in discussions as followed by Line 280-307, which are as below:
In general, recheck the manuscript and correct some typing errors.
Response 10: Thank you very much. We have rechecked the manuscript seriously.
Thank you again for your kind work! Please contact us if you have any queries about the manuscript!

Round 2
Reviewer 1 Report
Comments and Suggestions for Authors
The manuscript entitled "Genetic diversity of Cupressus gigantea W. C. Cheng & L. K. Fu by inter-simple sequence repeat” sounded in science and agriculture. The Founding of this study will be useful for C.gigantea conservation and breeding in future.
From R1, this manuscript was improved, but one point needs to be concerned as “The average of Nei’s gene diversity (H) of this study is 0.199”. This is rich genetic diversity? Please re-mention in conclusion.
Overall, I accept this manuscript to publication in Agronomy, after revision as recommend.
Best regards
Author Response
Dear Editors and Reviewers:
Thank you very much for your affirmation of our work and your constructive comments on current manuscript. We have carefully studied the comments on a point-by-point basis and revised our manuscript accordingly. The modifications we made are detailed in our replies below:
Reviewer: 1
Comments and Suggestions for Authors
The manuscript entitled "Genetic diversity of Cupressus gigantea W. C. Cheng & L. K. Fu by inter-simple sequence repeat” sounded in science and agriculture. The Founding of this study will be useful for C.gigantea conservation and breeding in future.
From R1, this manuscript was improved, but one point needs to be concerned as “The average of Nei’s gene diversity (H) of this study is 0.199”. This is rich genetic diversity? Please re-mention in conclusion.
Response : We thank the reviewer for pointing this viewpoint out. In the revised version, we have revised this inappropriate conclusion to say that “a large proportion of genetic diversity is present at the within-population level in C. gigantea”. Moreover, we added some content to the conclusion. They are followed:
Overall, I accept this manuscript to publication in Agronomy, after revision as recommend.
Thank you again for your kind work! Please contact us if you have any queries about the manuscript!
